# Spread Preference Annotation: Direct Preference Judgment for Efficient LLM Alignment

**Dongyoung Kim[1], Kimin Lee[1], Jinwoo Shin[1], Jaehyung Kim[2]**
[1]Korea Advanced Institute of Science and Technology , [2]Yonsei University
kingdy2002@kaist.ac.kr, jaehyungk@yonsei.ac.kr

## Abstract

Aligning large language models (LLMs) with human preferences becomes a key component to obtaining state-of-the-art performance, but it yields a huge cost to construct a large human-annotated preference dataset. To tackle this problem, we propose a new framework, **S**pread **P**reference **A**nnotation with direct preference judgment (SPA), that boosts the alignment of LLMs using only a very small amount of human-annotated preference data. Our key idea is leveraging the human prior knowledge within the small (seed) data and progressively improving the alignment of LLM, by iteratively generating the responses and learning from them with the self-annotated preference data. To be specific, we propose to derive the preference label from the logits of LLM to explicitly extract the model's inherent preference. Compared to the previous approaches using external reward models or implicit in-context learning, we observe that the proposed approach is significantly more effective. In addition, we introduce a noise-aware preference learning algorithm to mitigate the risk of low quality within generated preference data. Our experimental results demonstrate that the proposed framework significantly boosts the alignment of LLMs. For example, we achieve superior alignment performance on AlpacaEval 2.0 with only 3.3% of the ground-truth preference labels in the Ultrafeedback data compared to the cases using the entire data or state-of-the-art baselines.[1]

## 1 Introduction

Recently, large language models (LLMs) have made huge progress in various NLP tasks, leading to real-world applications that are used by millions of users, such as coding assistants and chatbots (Anthropic, 2024; OpenAI, 2022; Team et al., 2023). Aligning LLMs with human feedback, particularly through learning from human preferences, is widely considered a crucial technique for their success (Christiano et al., 2017; Lee et al., 2021; Ziegler et al., 2019). To enhance this alignment, various preference learning algorithms have been extensively explored (Ouyang et al., 2022; Rafailov et al., 2023). Despite these advancements, one of the remaining challenges is the reliance on large-scale human-annotated preference data. As the quality and quantity of the preference data are critical for the successful alignment of LLMs (Bai et al., 2022a; Cui et al., 2023), the huge cost to acquire such data inevitably presents significant obstacles.

To mitigate this challenge, engaging LLMs in constructing preference data and improving their alignment using these data has recently gained attention. For example, a representative way on this line is generating multiple responses for the input prompts, and then approximating human preference between them through LLM's predictions, often referred to as *LLM-as-judge* (Bai et al., 2022b; Yuan et al., 2024). However, these approaches are only effective when the given LLM is sufficiently large and well-aligned to mimic human preference via in-context learning. On the other hand, using an external reward model is considerable to substitute human preference annotation efficiently (Jiang et al., 2023b; Snorkel, 2024), but it is built on the availability of large human preference data and could also be ineffective if there is a distribution mismatch. Lastly, these approaches have a risk of potential labeling noise from LLMs, but this aspect has not been explored yet. Therefore, in this work, we aim to develop a method to effectively improve the alignment of LLM by overcoming these limitations but only relying on small human annotation.

---

[1]https://github.com/kingdy2002/SPA

Figure 1: **Illustration of the proposed SPA framework.** SPA progressively improves the alignment of LLMs by iterating (1) the generation of new preference data and (2) the preference learning on the constructed data with self-refinement. Technical details are presented in Section 4.

**Contribution.** We introduce a simple yet effective framework, coined SPA, to improve the alignment of LLMs with only a small amount of human-labeled preference data, by **S**preading **P**reference **A**nnotation via direct preference judgment. Our key idea is to progressively expand the knowledge of human preference within the small (seed) data, by iteratively generating the responses and learning from them through the self-annotated preference labels. Specifically, our technical contributions are three-fold as described in what follows. First, we judge the preference labels directly using the logits of LLM to explicitly extract the model's inherent preference. This approach is more effective than previous methods that rely on external reward models or implicit in-context learning. Second, we introduce a confidence-based refinement of preference labels to reduce the risk of noise in preference learning with generated data. Third, to further enhance the effectiveness of this refinement, we propose using a linearly extrapolated prediction between current and reference models; it approximates predictions of a more strongly aligned model, leading to better noise identification.

We demonstrate the effectiveness of the proposed SPA by aligning recent LLMs with small human-annotated preference data and evaluating their alignment on the commonly used benchmarks. For example, using only 3.3% of ground-truth preference in Ultrafeedback data (Cui et al., 2023) with the mistral-7b-0.1v SFT model (Jiang et al., 2023a), our framework achieves over 16.4% increase in AlpacaEval2.0 (Li et al., 2023a) win rate compared to the initial SFT model (see Figure 2). Additionally, the AlpacaEval 2.0 length-controlled win rate is improved from 7.58% to 15.39%, and MT-bench score (Zheng et al., 2023) increased from 6.38 to 6.94. Compared to preference judgment methods like LLM-as-judge (Zheng et al., 2023), and even strong reward models such as PairRM (Jiang et al., 2023b), which have recently shown state-of-art performance in AlpacaEval2.0 benchmark, our approach consistently outperforms them across all metrics. More interestingly, the proposed SPA successfully improves the alignment of various LLMs, even without the initial human preference data. These results demonstrate that our framework is highly competitive and practical for real-world applications.

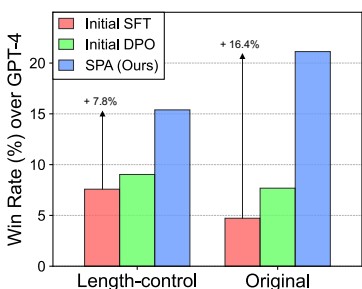

Figure 2: **Summary of main result.** Evaluation results on AlpacaEval 2.0 (Li et al., 2023a). Our framework significantly improves the alignment of LLMs, without additional human preference data. See detailed results in Section 5.

## 2 RELATED WORK

**Alignment of LLMs with human preference.** Learning from human preferences now serves as a core component for the state-of-the-art LLMs (Anthropic, 2024; OpenAI, 2023; Team et al., 2023; Touvron et al., 2023) for aligning their responses with users' intent and values (Ouyang et al., 2022; Ziegler et al., 2019). Arguably, one of the most popular frameworks is reinforcement learning with human preference (RLHF) (Christiano et al., 2017; Lee et al., 2021), which first trains the reward model, and then fine-tunes LLM to maximize that reward with KL divergence regularization to prevent the reward over-optimization of LLM. On the other hand, various preference learning algorithms have recently been proposed to fine-tune LLMs with human preference more efficiently (Ethayarajh et al.,

2024; Hong et al., 2024; Liu et al., 2023; Rafailov et al., 2023; Xu et al., 2023; Zhao et al., 2023; Meng et al., 2024). For example, Rafailov et al. (2023) proposes Direct Preference Optimization (DPO) which allows one to fine-tune LLMs without a separate reward modeling stage, by deriving the training objective mathematically equivalent to RLHF. Ethayarajh et al. (2024) further removes the reliance on pair-wise preference labels by formulating the objective based on a human utility model. However, these methods assume that large human-annotated preference data is available, which requires a huge data acquisition cost.

**Engagement of LLMs for constructing preference data.** For an efficient and scalable alignment procedure, engaging LLMs for preference dataset construction has recently received attention. One common approach involves generating multiple responses to input prompts from LLM, and using an LLM's predictions to approximate human preferences between them, a technique often referred to as *LLM-as-judge* (Bai et al., 2022a; Yuan et al., 2024). However, this method is effective only when the LLM is sufficiently large and well-aligned to mimic human preferences through in-context learning. Alternatively, employing an external reward model can efficiently replace human preference judgment (Jiang et al., 2023b; Snorkel, 2024), but this approach relies on the availability of extensive human preference data to pre-train reward model and may be ineffective if there is a distribution mismatch. Some concurrent works (Rosset et al., 2024; Snorkel, 2024; Wu et al., 2024; Xiong et al., 2024) have proposed the alignment procedure with iterative data expansion and preference learning. However, they use the external reward model or stronger LLM for the preference judgment. In contrast, we only utilize the intrinsic knowledge of training LLM for new data expansion and preference learning.

## 3 PRELIMINARIES

Let us denote LLM as $\pi_\theta$, which generates an output sequence (*e.g.*, response) $y$ for a given input sequence (*e.g.*, prompt) $x$, *i.e.*, $y \sim \pi_\theta(\cdot|x)$. Then, our goal is to make $\pi_\theta$ provide human-aligned responses to various input prompts. To this end, we consider the popular framework of preference learning, which optimizes $\pi_\theta$ to learn human preferences between two different responses (Christiano et al., 2017; Lee et al., 2021; Ouyang et al., 2022). Specifically, we assume that the preference dataset $\mathcal{D} = \{(x, y_l, y_w)\}$ is available which consists of the triplets of input prompt $x$, preferred response $y_w$, and dispreferred response $y_l$. Here, the preference labels were annotated by a ground truth annotator, that is usually a human expert.

**Reward modeling and RL fine-tuning.** Since a pairwise preference between $y_w$ and $y_l$ is hard to model directly, one of the common practices is introducing reward function $r(x, y)$ and modeling the preference based on this using the Bradley-Terry model (Bradley & Terry, 1952):

$$p(y_w \succ y_l \mid x) = \frac{\exp\left(r(x, y_w)\right)}{\exp\left(r(x, y_w)\right) + \exp\left(r(x, y_l)\right)}. \tag{1}$$

From this formulation, one can introduce a parametrized reward model $r_\phi(x, y)$ by estimating its parameters with the maximum-likelihood objective:

$$\mathcal{L}_R(r_\phi) = -\mathbb{E}_{(x, y_w, y_l) \sim \mathcal{D}} \left[\log \sigma\left(r_\phi(x, y_w) - r_\phi(x, y_l)\right)\right]. \tag{2}$$

where $\sigma$ is a sigmoid function. After this reward modeling procedure, one could improve the alignment of LLM $\pi_\theta$ by optimizing it to maximize the reward captured by $r_\phi$. Here, KL-distance from the reference model $\pi_{\text{ref}}$ is usually incorporated as a regularization to prevent the reward over-optimization of $\pi_\theta$, with a hyper-parameter $\beta > 0$ (Ouyang et al., 2022; Ziegler et al., 2019):[2]

$$\mathcal{L}_{\text{RLHF}}(\pi_\theta) = -\mathbb{E}_{y \sim \pi_\theta, x \sim \rho} \left[r_\phi(x, y)\right] + \beta \text{D}_{\text{KL}}\left(\pi_\theta(y|x) \parallel \pi_{\text{ref}}(y|x)\right). \tag{3}$$

**Direct preference modeling and optimization.** Rafailov et al. (2023) propose an alternative approach to align LLM $\pi_\theta$ with the preference dataset $\mathcal{D}$, which is called Direct Preference Optimization (DPO). DPO integrates a two-step alignment procedure with reward modeling and RL fine-tuning into a single unified fine-tuning procedure. Specifically, the optimal reward function is derived from the

---

[2] $\pi_{\text{ref}}$ is usually initialized with supervised fine-tuned (SFT) LLM (Chung et al., 2024; Wei et al., 2022a). Also, $\pi_\theta$ is initialized with $\pi_{\text{ref}}$.

RLHF objective (Eq. 3), with the target LLM $\pi_\theta$ and the reference model $\pi_{\text{ref}}$ (Go et al., 2023; Peng et al., 2019; Peters & Schaal, 2007).

$$r(x, y) = \beta \log \frac{\pi_\theta(y \mid x)}{\pi_{\text{ref}}(y \mid x)} + \beta \log Z(x), \text{ where } Z(x) = \sum_y \pi_{\text{ref}}(y \mid x) \exp\left(\frac{1}{\beta} r(x, y)\right). \quad (4)$$

Then, the preference between two responses could be measured using this reward derivation, and $\pi_\theta$ is optimized to maximize this preference of $y_w$ over $y_l$ using the preference dataset $\mathcal{D}$.

$$p_\theta(y_w \succ y_l | x) = \sigma\left(\beta \log \frac{\pi_\theta(y_w|x)}{\pi_{\text{ref}}(y_w|x)} - \beta \log \frac{\pi_\theta(y_l|x)}{\pi_{\text{ref}}(y_l|x)}\right). \quad (5)$$

$$\mathcal{L}_{\text{DPO}}(\pi_\theta) = \mathbb{E}_{(x,y_w,y_l)\sim\mathcal{D}}\left[-\log p_\theta(y_w \succ y_l | x)\right]. \quad (6)$$

## 4   SPA: SPREAD PREFERENCE ANNOTATION TO BOOST ALIGNMENT OF LLMS

**Overview.** In this section, we present SPA: **S**pread **P**reference **A**nnotation via direct preference judgment to align LLMs while mitigating the huge cost for preference dataset construction. Our main idea is to fully exploit the human prior knowledge within the small (seed) data, and progressively update LLM to improve the alignment. To be specific, SPA iterates two steps: (1) data expansion with self-generated preference (Section 4.1) and (2) fine-tuning LLM with self-refined preference learning (Section 4.2). See Figure 1 for the overview.

**Initial stage**. We assume that a small (seed) preference dataset $D_0$ and an initial LLM $\pi_{\text{init}}$ are given. Here, following the common practice (Ouyang et al., 2022; Rafailov et al., 2023; Ziegler et al., 2019), we use $\pi_{\text{init}}$ which has been supervised fine-tuned (SFT) LLM on the instruction dataset (Chung et al., 2024; Wei et al., 2022a), but not aligned with human preference yet. Then, we first obtain weakly aligned LLM $\pi_0$ by fine-tuning $\pi_{\text{init}}$ on $D_0$ using DPO (Rafailov et al., 2023) (Eq. 6). We adopt DPO among various preference learning methods due to its simplicity and effectiveness.

### 4.1   DIRECT PREFERENCE JUDGMENT TO ALIGN LLMS WITH SELF-GENERATED DATA

For the $i$-th iteration ($i = 1, \dots$), we assume that the new prompt set $X_i = \{x\}$ is available, *i.e.*, $X_i \cap X_j = \emptyset$ for all $j = 0, \dots, i-1$.[3] From $X_i$, we construct $i$-th artificial preference dataset $\mathcal{D}_i = \{(x, y_l, y_w) | x \in X_i\}$, by using LLM's intrinsic generation and reward modeling capabilities. Specifically, for each input prompt $x \in X_i$, we sample two responses $y_1$ and $y_2$ from $\pi_{i-1}$, *i.e.*, $y_1, y_2 \sim \pi_{i-1}(x)$ where $\pi_{i-1}$ is the resulting model from the previous iteration. Then, using the reward captured with $\pi_{i-1}$ and $\pi_{\text{init}}$ (Eq. 4), we measure the preference of $\pi_{i-1}$ between $y_1$ and $y_2$:

$$p_{i-1}(y_1 \succ y_2 | x) = \sigma\left(\beta \log \frac{\pi_{i-1}(y_1|x)}{\pi_{\text{init}}(y_1|x)} - \beta \log \frac{\pi_{i-1}(y_2|x)}{\pi_{\text{init}}(y_2|x)}\right). \quad (7)$$

Then, we directly judge the preference label as below and construct $\mathcal{D}_i$ through this:

$$(y_w, y_l) = (y_1, y_2) \text{ if } p_{i-1}(y_1 \succ y_2 | x) > 0.5 \text{ else } (y_w, y_l) = (y_2, y_1). \quad (8)$$

### 4.2   SELF-REFINEMENT OF GENERATED PREFERENCE DATA FOR EFFECTIVE LEARNING

After the construction of $\mathcal{D}_i$, we conduct $i$-th preference learning by fine-tuning $\pi_\theta$, which is initialized by $\pi_{i-1}$, using DPO (here, we also use $\pi_{i-1}$ as $\pi_{\text{ref}}$ in Eq. 6). Learning the self-generated preference data $\mathcal{D}_i$ could improve the alignment by effectively spreading the human preference prior from $\mathcal{D}_0$ using the power of LLM. However, it also has a risk of the potential labeling noise which could occur from the distribution shift with $X_i$ or insufficient reward modeling with $\pi_{i-1}$. Therefore, we further propose an improved preference learning method by introducing a novel denoising technique: *self-refinement* of preference labels with *de-coupled noise detection*.

---

[3] $X_0 = \{x | (x, y_l, y_w) \in \mathcal{D}_0\}$

---

**Algorithm 1** SPA algorithm

---

**Input:** initial LLM $\pi_{\text{init}}$, seed preference dataset $\mathcal{D}_0$, number of improving iterations $T$, new prompt sets $\{X_i\}_{i=1}^T$,

---

Obtaining an initial weakly aligned model $\pi_0$ using DPO with $\pi_{\text{init}}$ and $\mathcal{D}_0$ (Eq. 6)
**for** $t = 1$ **to** $T$ **do**
    Synthesizing preference data $\mathcal{D}_t$ with $\pi_{t-1}$ and $X_t$ (Eq. 7 and 8)
    Initialization of training and reference models $\pi_\theta \leftarrow \pi_{t-1}$, $\pi_{\text{ref}} \leftarrow \pi_{t-1}$
    **for** mini-batch $B \sim \mathcal{D}_t$ **do**
        $z_{\widetilde{\theta}} \leftarrow$ De-coupled noise detection for $B$ from $\pi_\theta, \pi_{\text{ref}}, X_t$ (Eq. 11 and 12)
        Calculate training loss $\mathcal{L}_{\text{rf}}$ with refined preference labels using $z_{\widetilde{\theta}}$ and $\pi_\theta$ (Eq. 10)
        Update model parameter: $\theta \leftarrow \theta - \eta\nabla_\theta \mathcal{L}_{\text{rf}}$
    **end for**
    Initializing next iteration model $\pi_t$ with the updated parameters $\theta$
**end for**
**return** $\pi_T$

---

**Self-refinement of preference label**: Our key intuition is that one can view the derived preference (Eq. 5) can be viewed as the confidence of the currently training LLM $\pi_\theta$ for the labels assigned by $\pi_{i-1}$. Then, $\pi_\theta$ would exhibit lower confidence if the given pair of responses is uncertain to answer, indicating a higher probability of labeling noise. Notably, we also remark that confidence is one of the most popular metrics in the noisy label learning literature (Han et al., 2018; Reed et al., 2014; Sohn et al., 2020). Under this intuition, we first identify the $K\%$ least confident samples:

$$z_\theta = 1 \text{ if } p_\theta(y_w \succ y_l|x) < \tau \text{ else } z_\theta = 0, \tag{9}$$

where $\tau$ is the confidence of $K$ percentile sample of $\mathcal{D}_i$. Then, with this (potentially) noise identification label $z_\theta$, we refine the assigned preference label using label smoothing (Müller et al., 2019), to train $\pi_\theta$ less confidently when the risk of label noise is high (*i.e.*, $z_\theta = 1$):

$$\mathcal{L}_{\text{rf}}(\pi_\theta) = \mathbb{E}_{(x,y_w,y_l)\sim\mathcal{D}_i}\left[-\big((1 - \alpha * z_\theta)\log p_\theta(y_w \succ y_l|x) + \alpha * z_\theta \log p_\theta(y_l \succ y_w|x)\big)\right], \tag{10}$$

where $\alpha$ is a hyper-parameter. Then, we train $\pi_\theta$ using $\mathcal{L}_{\text{rf}}(\pi_\theta)$ instead of naive DPO (Eq. 6).

**De-coupled noise preference detection**: While learning with the refined preference label reduces the risk of learning $\pi_\theta$ the noisy preference, its effectiveness could be limited as the model $\pi_\theta$ for noise detection originated from the label generation model $\pi_{i-1}$. Therefore, to further improve the effectiveness of our preference label refinement framework, we introduce the de-coupled noise detection (Han et al., 2018; Li et al., 2020) technique for LLM alignment. Specifically, we identify the preference noise by mimicking the preference prediction of a more strongly aligned LLM $\pi_{\widetilde{\theta}}$: [4]

$$z_{\widetilde{\theta}} = 1 \text{ if } p_{\widetilde{\theta}}(y_w \succ y_l|x) < \tau \text{ else } z_{\widetilde{\theta}} = 0. \tag{11}$$

With this de-coupled identification, $\pi_\theta$ is trained with refined preference labels via Eq. 10 , *i.e.*, $z_{\widetilde{\theta}}$ is used to substitute $z_\theta$ in Eq. 10. Here, we obtain the prediction of $\pi_{\widetilde{\theta}}$ by approximating its logit $h_{\widetilde{\theta}}$ through the linear combination of the logits of $\pi_\theta$ and $\pi_{\text{ref}}$. [5] It is motivated by the recent work (Liu et al., 2024) that shows the aligned models via RLHF with varying $\beta$ are geometric mixtures of a reference model and a single aligned model:

$$h_{\widetilde{\theta}}(x, y_{1:t-1}) = (1 + \lambda) * h_\theta(x, y_{1:t-1}) - \lambda * h_{\text{ref}}(x, y_{1:t-1}), \tag{12}$$

where $\lambda > 0$ is a hyper-parameter and $y_{1:t-1}$ indicates the output sequence before $t$-th output.

We remark that this de-coupled noise identification by approximating $p_{\widetilde{\theta}}(y_w \succ y_l|x)$ *does not require additional computations* compared to DPO, since the required measurements $h_\theta$ and $h_{\text{ref}}$ are obtained during the calculation of the original DPO objective (Eq. 6). Therefore, SPA only requires a few lines of additional code to the original DPO codebase. We present full procedure of SPA in Algorithm 1.

---

[4]With $\lambda$ in Eq. 12, $\pi_{\widetilde{\theta}}$ is equivalent to model trained with $(1 + \lambda)$ times smaller KL term than $\pi_\theta$ via Eq. 3.
[5]When $\pi_\theta(\cdot|x) := \text{Softmax}\big(h_\theta(x)\big)$, we refer $h_\theta(x)$ as the logit of LLM $\pi_\theta$ for the given input $x$.

## 5 EXPERIMENTS

In this section, we present our experimental results to answer the following question:

- ○ Does SPA improve the alignment of LLMs only using a small amount of human-labeled preference data? (Table 1, Figure 4)
- ○ Does the proposed method outperform other preference labeling methods? (Table 2, Figure 3)
- ○ Is SPA generalizable across various choices of seed data and types of LLMs? (Tables 3,4,5)
- ○ What is the effect of each component in SPA? (Tables 6,7)

### 5.1 EXPERIMENTAL SETUPS

**Models.** When there are no specific mentions, our experiments were conducted using the supervised fine-tuned Mistral-7b-0.1 model (Jiang et al., 2023a), as the initial model $\pi_{\text{init}}$ in Section 4. Specifically, we use the open-sourced model[6] that follows the recipe of Zephyr (Tunstall et al., 2023) and fine-tuned on the instructions of Ultrachat (Ding et al., 2023). More details are in Appendix B.

**Baselines.** To evaluate the effectiveness of the proposed preference judgment method (Eq. 7), we compare it with other preference judgment methods. Specifically, we consider the baselines that train the model via Iterative DPO (Snorkel, 2024; Xu et al., 2023), which iteratively generate preference data and update the model, using LLM-as-judge (Bai et al., 2022b; Zheng et al., 2023) (*i.e.*, in-context learning) or an external powerful reward model (PairRM (Jiang et al., 2023b)) for the preference judgment. Notably, these approaches are the same in the case of changing the judgment method and removing self-refinement in SPA. Details are presented in Appendix B.

**Datasets.** For the preference learning dataset, we utilized UltraFeedback (Cui et al., 2023), following the previous works (Snorkel, 2024; Rosset et al., 2024).[7] To be specific, from this dataset, we first construct the seed data, consisting of 2K samples (3.3% of 60K) with prompts, responses, and ground truth preference labels. We refer the ground-truth preference label provided by the UltraFeedback as *gold label* in Tables 1 and 5. Then, the remaining samples are divided into subsets of 8K, 20K, and 30K samples, leaving only the prompts. These subsets were used as the prompt sets for the iteration stages 1, 2, and 3, respectively. Only for the experiments in Table 3, the size of seed data is changed.

**Evaluations.** Following the common practice in LLM alignment, we mainly evaluate each model our evaluations using (1) AlpacaEval 2.0 (Dubois et al., 2023; 2024; Li et al., 2023a). AlpacaEval 2.0 approximately evaluates human preference for instruction following. Using 805 instructions from various datasets, the evaluation is conducted by comparing the response of GPT-4 (OpenAI, 2023) and the testing model to measure win rates. To mitigate the length bias of LLM's preference (Wang et al., 2023b; Zheng et al., 2023), both original and length-controlled (LC) win rates are simultaneously measured. LC win rate is an adjusted win rate by neutralizing the effect of response length to focus on quality, using a separately trained regression model (Dubois et al., 2024). We also evaluate trained LLMs using (2) MT-Bench (Zheng et al., 2023) to assess different aspects of LLMs. Namely, MT-Bench evaluates a chatbot's overall abilities across multiple categories related to key LLM capabilities such as math, coding, roleplay, writing, etc. The evaluation is conducted by scoring responses to multi-turn questions using GPT-4. These benchmarks also provide a thorough evaluation of LLMs' alignment with human preferences and their overall effectiveness in practical applications.

**Implementation details.** After the initialization stage, we conduct three rounds of data expansion with self-generated preference data. For data expansion, we sampled 2 responses independently per each prompt with a temperature of 0.7. Then, using the SFT model as the reference model, we assign the preference label (Eq. 7). The initial DPO training to obtain $\pi_0$ was conducted for 3 epochs on the seed dataset. Training on each subsequent iteration was carried out for 1 epoch. For the hyper-parameter $\beta$ of DPO, we used a fixed value of $\beta = 0.1$. The batch size was set to 32, and the learning rate was $5 \times 10^{-7}$. We employed AdamW optimizer and a cosine learning rate scheduler with a warm-up phase corresponding to 10% of the total training steps. For the hyper-parameters $\alpha$ and $K\%$ for SPA, we used fixed values of $\alpha = 0.1$ and $K = 10$. Additionally, a warm-up phase was included in the denoising stage, with denoising activated after 20% of the total training steps had been completed. Regarding the hyper-parameters $\lambda$ for de-coupled noise detection, we utilized the progressively reduced values of 1/2, 1/4, and 1/8 for iterations 1, 2, and 3, respectively.

---

[6]`alignment-handbook/zephyr-7b-sft-full`
[7]`"argilla/ultrafeedback-binarized-preferences-cleaned"`

Table 1: **Main results.** Evaluation results on AlpacaEval 2.0 and MT-Bench with different variants of Mistral-7B-v0.1. The best scores are highlighted with **bold**.

| Models | Gold Label (%) | AlpacaEval 2.0 | | MT-Bench |
| | | Len-control. Win Rate (%) | Win Rate vs. GPT-4 (%) | Avg. Score (0-10) |
| --- | --- | --- | --- | --- |
| Mistral-7B-v0.1 | - | 0.17 | 0.50 | 3.25 |
| Zephyr-7b-$\beta$ | 100 | 11.75 | 10.03 | 6.87 |
| SFT | - | 7.58 | 4.72 | 6.34 |
| DPO | 3.3 | 9.03 | 7.68 | 6.81 |
| SPA (Ours) | 3.3 | **15.39** | **21.13** | **6.94** |

Table 2: **Comparison with baselines for preference judgment.** Evaluation results on AlpacaEval 2.0 and MT-Bench with iteratively trained models (from SFT model) under different preference judgment methods. The best scores are highlighted with **bold**.

| Methods | External Model | AlpacaEval 2.0 | | MT-Bench |
| | | Len-control. Win Rate (%) | Win Rate vs. GPT-4 (%) | Avg. Score (0-10) |
| --- | --- | --- | --- | --- |
| Iterative DPO (PairRM) | ✓ | 11.87 | 9.46 | **6.98** |
| Iterative DPO (LLM-as-judge) | ✗ | 9.28 | 9.18 | 6.67 |
| SPA (Ours) | ✗ | **15.39** | **21.13** | 6.94 |

## 5.2 MAIN RESULTS

After completing 3 iterations of data expansion and fine-tuning via SPA, the trained model achieved a 21.13% win rate against GPT-4 on the AlpacaEval 2.0 benchmark, as presented in Table 1. This represents a significant improvement compared to the 7.68% (7.68% → 21.13%) win rate achieved when using only 3.3% of labeled data with the standard DPO training, while the length-control win rate is also improved. (9.03% → 15.39%). In addition, SPA achieved a score of 6.94 on the MT-Bench, clearly outperforming the model trained with DPO (6.81) on the same amount of 3.3% gold labeling data. More interestingly, our framework achieved superior performance in both win rate (10.03% vs 21.13%) and length-control win rate (11.75% vs 15.39%), compared to Zephyr-7b-$\beta$ which uses same base model (Mistral-7B-0.1v) and SFT dataset but uses significantly larger labeled preference data, *i.e.*, 100% of UltraFeedback dataset (v.s. 3.3% for SPA). These significant improvements in both win rates clearly affirm the overall enhancement in performance from SPA.

Next, in Table 2, we present additional experimental results to validate the proposed preference judgment method. Namely, three experiments in Table 2 can be viewed as the Iterative DPO variants with different preference judgment methods. One can observe that SPA showed significantly better performance compared to other methods. Specifically, SPA achieved a win rate of 21.13% against GPT-4 on AlpacaEval 2.0, compared to 9.46% for the baseline with an external reward model, PairRM. In terms of length control win rate, SPA achieved 15.39%, surpassing the reward model's 11.84%. Here, we conjecture that the reason why the Iterative DPO training with the proposed direct preference judgment method (using training LLM) outperforms the case with inferred labels from the external reward model is related to the distribution shift. As the iteration is increased, the distribution of the generated data with LLM is more shifted from the distribution of the seed preference data. Then, the effectiveness of the external reward model inevitably decreases, as the

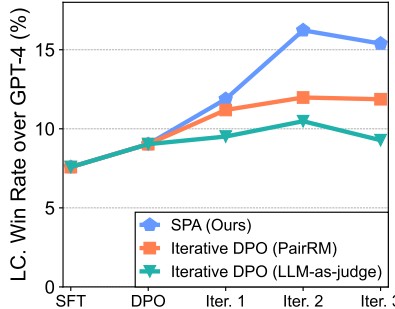

Figure 3: **Improvements during iterations.** Length control (LC.) win rate (%) measured by AlpacaEval 2.0 is consistently improved by SPA and it outperforms other baselines.

Table 3: **Different number of seed data.** Evaluation results on AlpacaEval 2.0 with Mistral-7B-v0.1 trained with DPO and SPA under the different number of seed ground-truth preference labels.

| Methods | Used Ground-truth Preference Data | | | |
|---|---|---|---|---|
| | 0.8% | 1.7% | 3.3% | 10% |
| DPO: LC Win Rate (%) | 7.85 | 7.68 | 9.03 | 11.37 |
| DPO: Win Rate (%) | 5.53 | 5.49 | 7.68 | 9.32 |
| SPA: LC Win Rate (%) | 10.36 | 12.36 | 16.23 | 18.52 |
| SPA: Win Rate (%) | 11.34 | 13.72 | 19.94 | 23.79 |

reward model is fixed while the generated data is increasingly distant from its training distribution. In contrast, SPA generates the preference label using the intrinsic reward model that is continuously updated for each iteration. Therefore, it less suffers from the distribution shift during the iterations, and hence could be more effective for iterative training. Regarding this, we remark on the results in Figure 3; at iteration 1, the effectiveness of both approaches is not much different. However, the gap is significantly widened at iteration 2, and it empirically supports the above rationale.

On the other hand, the in-context learning approach (LLM-as-judge) shows a similar win rate compared to PairRM, but falls short in length control win rate (11.87% vs 9.28%), showing the limitations of the LLM-as-judge approach. Overall, the results reveal the superiority of our direct preference judgment over other judgment methods. Also, this superiority is consistently observed through the iterations, as shown in Figure 3.

## 5.3 MORE ANALYSES

In this section, we conduct additional analyses of SPA by comparing the results on AlpacaEval 2.0. More comparisons on the MT-Bench and the additional experiments are presented in the Appendix.

**Generalization across different numbers of seed data.** Previously, we conducted the experiment by assuming that only a limited number of human preference data is initially given, *e.g.*, 3.3% of UltraFeed-back dataset. However, the effectiveness of SPA does not depend on the size of the seed preference dataset and we validate this with the additional experiments. First, we conduct the experiments by varying the portion of the seed ground-truth preference data. Specifically, to use the fixed input prompt datasets for each iteration, we consider the following portions for the experiments: [0.8%, 1.7%, 10%]. Table 3 shows the results on AlpacaEval 2.0 with Mistral-7B-v0.1 after 2 iterations of training with SPA, including the original experiments with 3.3% seed preference data. Here, one can observe that the alignment performance under DPO and SPA is improved with the increased seed data, and SPA consistently outperforms DPO which demonstrates the robustness of SPA regarding the size of seed preference data.

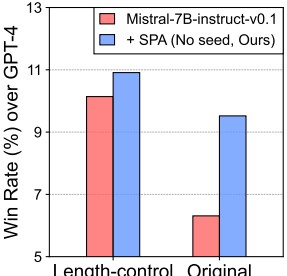

Figure 4: **Improvements without seed data.** Evaluation results on AlpacaEval 2.0 with Mistral-7B-instruct-v0.1 and SPA with no seed preference data.

We further evaluated the feasibility of using SPA even *without seed preference data*. Namely, we want to answer whether LLM can derive explicit human preference between responses, by leveraging their intrinsic knowledge learned about humans, during the previous training, such as pre-training or supervised instruction tuning (SFT). For this experiment, we used the Mistral-7b-instruct-0.1v (Jiang et al., 2023a) as the initial model (*i.e.*, $\pi_0$) and the Mistral-7b-0.1v-base as the reference model (*i.e.*, $\pi_{\text{init}}$) (see the initial setup in Section 4). This setup allows us to demonstrate that our framework can function effectively even in the absence of seed preference data, when the model is sufficiently fine-tuned with iterative data expansion and learning through self-refinement. As shown in Figure 4, the win rate increased from 6.31% to 9.79%, and the length-control win rate improved from 10.14% to 11.59%. This result indicates that SPA can leverage the internal information of LLMs to be aligned with human preference even without seed data.

Table 4: **Different initial seeds.** Evaluation results on AlpacaEval 2.0 with different variants of Mistral-7B-v0.1 under the different sampling of the initial seed preference data.

| Methods | 1st Seed Data | 2nd Seed Data | 3rd Seed Data | Average | Variance |
|---|---|---|---|---|---|
| DPO: LC Win Rate (%) | 9.03 | 8.74 | 9.54 | 9.10 | 0.16 |
| DPO: Win Rate (%) | 7.68 | 7.17 | 7.59 | 7.48 | 0.07 |
| SPA (Ours): LC Win Rate (%) | 16.23 | 13.77 | 16.38 | 15.46 | 2.10 |
| SPA (Ours): Win Rate (%) | 19.94 | 20.06 | 19.74 | 19.91 | 0.03 |

Table 5: **Compatibility across various LLMs.** Evaluation results on AlpacaEval 2.0 with different training methods (SFT, DPO, and SPA) across various types of LLMs (Phi-2-2.7B, LLaMA-3-8B, and Phi-3-14B). The best scores are highlighted with **bold**.

| Methods | Gold Label (%) | Phi-2-2.7B | | LLaMA-3-8B-Instruct | | Phi-3-14B-Instruct | |
|---|---|---|---|---|---|---|---|
| | | Len-control. Win Rate (%) | Win Rate vs. GPT-4 (%) | Len-control. Win Rate (%) | Win Rate vs. GPT-4 (%) | Len-control. Win Rate (%) | Win Rate vs. GPT-4 (%) |
| SFT | - | 5.88 | 3.78 | 21.40 | 21.71 | 26.51 | 21.41 |
| DPO | 3.3 | 7.02 | 5.67 | 24.17 | 25.39 | 27.70 | 22.12 |
| SPA (Ours) | 3.3 | **9.10** | **9.43** | **25.03** | **34.84** | **28.77** | **24.14** |

**Variance with different initial seed dataset.** In addition, we conduct experiments to check the sensitivity of SPA with the initial seed preference dataset by varying them with different random sampling. The results after 2 iterations of training with SPA are presented in Table 4. Here, one can observe that the proposed SPA consistently improves the alignment performance regardless of the given seed data, and the variance between them is not significant, especially in the case of a normal win rate. While ours exhibits a relatively high variance for length-controlled (LC) win rate, its lowest confidence interval value (13.36 %) is certainly higher than the value of the strongest baseline (11.98 %) which confirms the effectiveness of our method.

**Compatibility with different models.** Next, to verify the compatibility of our framework across various LLMs, we conducted experiments using three different LLMs: Phi-2-2.7B (Li et al., 2023b), LLaMA3-8B (Dubey et al., 2024), and Phi-3-14B. Specifically, we conducted experiments based on their supervised fine-tuned versions; for Phi-2, we used the model that has been fine-tuned on the UltraChat dataset like Mistral.[8] For LLaMA-3[9] and Phi-3[10], we used the generally fine-tuned models as there are no models that have been fine-tuned on the UltraChat dataset. Here, most of the experimental setups for these experiments are maintained, and the slightly adjusted setups are detailed in Appendix B.3. As shown in Table 5, the experimental results showed that applying SPA to various LLMs yields consistent improvements in the performance. For example, the win rate improved from 5.67% to 9.43%, and the length control win rate increased from 7.02% to 9.1%, in the case of Phi-2 after being trained with SPA compared to DPO. These results demonstrate that the effectiveness of SPA is not limited to the specific LLMs and is generalized across various LLMs.

**Ablation study.** To evaluate the impact of the self-refinement components, we conducted ablation experiments by excluding both self-refinement (SR) and decoupled noise detection (DND) from the existing framework. The results are presented in Table 6. With self-refinement without decoupled noise detection (Eq. 10), we observed a slight performance improvement, with the win rate against GPT-4 marginally increasing from 19.91% to 19.94%, and the length control win rate rising from 14.41% to 14.7%. But, when the decoupled noise detection is incorporated into the self-refinement (Eq. 11), we observed significant improvements, with the win rate increasing from 19.91% to 21.13% and the length control win rate improving from 14.41% to 15.39%. Also, these results confirm that the self-refinement component is a crucial factor in enhancing performance, contributing to both higher win rates and better length control.

**Additional analysis with judgment methods.** In Table 7, we further analyzed the impact of the reference model in the preference judgment process in Eq. 7. This analysis was conducted during

---

[8] `lole25/phi-2-sft-ultrachat-full`
[9] `https://huggingface.co/meta-llama/Meta-Llama-3-8B-Instruct`
[10] `https://huggingface.co/microsoft/Phi-3-medium-4k-instruct`

Table 6: **Ablation study.** Evaluation results on AlpacaEval 2.0 with iteratively trained models (from SFT) under different methodological configurations of SPA. DE, SR, DND are abbreviations of data expansion, self-refinement, and de-coupled noise detection, respectively. The best scores are highlighted with **bold**.

| Methods | DE | SR | DND | AlpacaEval 2.0 | |
| --- | --- | --- | --- | --- | --- |
| | | | | Len-control. Win Rate (%) | Win Rate vs. GPT-4 (%) |
| SFT | - | - | - | 7.58 | 4.72 |
| DPO | - | - | - | 9.03 | 7.68 |
| SPA (Ours) | ✓ | ✗ | ✗ | 14.41 | 19.91 |
| | ✓ | ✓ | ✗ | 14.7 | 19.94 |
| | ✓ | ✓ | ✓ | **15.39** | **21.13** |

Table 7: **Additional analyses.** Evaluation results on AlpacaEval 2.0 with models that fine-tuned with different judgment methods, from the resulting model of 1st iteration of SPA.

| Models | AlpacaEval 2.0 | |
| --- | --- | --- |
| | Len-control. Win Rate (%) | Win Rate vs. GPT-4 (%) |
| SPA after iteration 1 | 10.57 | 11.89 |
| Eq. 7 with initial SFT model (Ours) | 15.08 | 19.56 |
| Eq. 7 with previous model | 13.73 | 17.66 |
| Judgment with PairRM | 13.57 | 13.72 |
| Judgment without reference model | 12.83 | 12.35 |

the transition from iteration 1 to iteration 2, where the most significant performance changes were observed (*i.e.*, we fine-tune from the resulting model of iteration 1). To isolate and compare the effect of judgment methods, we followed the setup in Table 2 and so excluded the influence of the self-refinement component. Then, we experimented with three setups by varying the judgment method using (1) the current policy from the previous iteration as the reference model, (2) performing judgment without any reference model, and (3) using the PairRM for judgment.

The results are presented in Table 7. Here, the experimental results demonstrated that the method used in SPA, where the SFT model was utilized as the reference model for preference judgment, achieved the highest performance increase. Specifically, using the model from the previous iteration as the reference model showed lower performance, with a relatively larger decrease in the length control win rate (15.08% vs 13.73%) compared to the win rate (19.56% vs 17.66%). Despite these decreases, it still outperforms using PairRM. These results may imply the importance of judging the preference through the training LLM rather than the external model, as it is less suffering from the distribution mismatch. However, without reference model (*i.e.*, only using the likelihood of the current model), the performance increase was the lowest compared to all other cases. These findings underscore the substantial impact of the choice of proper judgment method and reference model.

## 6 CONCLUSION

In this paper, we proposed SPA, a method that can efficiently improve the alignment of LLMs using minimal human-labeled preference data. Our main contributions include the development of an effective data expansion method with the direct preference judgment method and a preference learning algorithm with the self-refinement of (potentially) noise preference. We demonstrate the effectiveness of SPA by fine-tuning the recent LLMs with the various setups, and observing the significant improvements when evaluating them on the commonly used benchmarks, AlpacaEval 2.0 and MT-Bench. We expect SPA to make significant contributions to future research and practical applications, especially when the human-labeled preference is hard to collect. Limitations and societal impacts are further discussed in Appendix A.

## REPRODUCIBILITY STATEMENT

For the reproducibility of our results, we have provided a detailed description of our methods and experimental setups in Section 5.1 and Appendix B. We also confirmed the robustness of our results through the experiment (Table 4). In addition, to further facilitate the reproduction, we will release our codes and the checkpoints for the trained models.

## ACKNOWLEDGMENTS

This work was mainly supported by Institute of Information & communications Technology Planning & Evaluation (IITP) grant funded by the Korea government (MSIT) (No.2021-0-02068, Artificial Intelligence Innovation Hub, 50%; RS-2022-II220959, Few-shot Learning of Casual Inference in Vision and Language for Decision Making, 50%).

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

# A LIMITATION AND SOCIETAL IMPACT

## A.1 LIMITATION AND FUTURE WORK

In the experiments, SPA has shown the tendency to increase the responses' length (please see Appendix D for the relevant results and discussions). We demonstrated that the improvement by SPA is not a simple result of such length increase, by observing the increase of win rate under a length length-controlled setup or MT-bench. However, depending on the user, this behavior could be dispreferred. In this sense, focusing on mitigating this bias during the self-improving alignment will be an interesting future direction, and can enhance the robustness and generalizability of SPA across more diverse scenarios.

## A.2 SOCIETAL IMPACT

SPA enables efficient human preference learning, allowing for cost-effective training of models in data-scarce or domain-specific areas. Our framework supports alignment learning in various fields, including multilingual language learning and preferences beyond human helpfulness. Consequently, it could contribute to facilitating the widespread adoption of LLM technology across diverse sectors. By lowering the barriers to alignment learning, SPA makes it more accessible to a broader audience. However, the widespread availability of this technology also brings potential risks. The reduced cost of training models could enable malicious actors to misuse the technology, leading to societal issues. Therefore, it is crucial to implement ethical considerations and safety measures when deploying SPA technology to mitigate these risks.

# B MORE DETAILS OF EXPERIMENTAL SETUPS

## B.1 SFT MODEL SETUP

**Mistral.** For supervised fine-tuning, Ultrachat dataset (Ding et al., 2023) is used[11], batch size was set 128, total epoch was 1, and the learning rate was $2 \times 10^{-5}$. It employed Adam optimizer (Kingma & Ba, 2015) and a cosine learning rate scheduler with a warm-up phase corresponding to 10% of the total training steps.

**Phi-2.** For supervised fine-tuning, Ultrachat dataset is used, batch size was set 64, total epoch was 3, and the learning rate was $2 \times 10^{-5}$. It employed Adam optimizer and a cosine learning rate scheduler with a warm-up phase corresponding to 10% of the total training steps.

**LLaMA-3 and Phi-3.** As described in Section 5.3, we use the generically instruct-tuned versions for both LLaMA-3-8B and Phi-3-14B, as there are no SFT models tuned on Ultrachat dataset.

## B.2 BASELINES EXPERIMENT SETUP

**Zephyr-7b-$\beta$.** We implemented Zephyr-7b-$\beta$ (Tunstall et al., 2023), which is compared in Table 1, according to recipes. Our Zephyr-7b-$\beta$ was trained using the same pre-trained model (mistral-7b-0.1v (Jiang et al., 2023a)) and the same SFT data (Ultrachat (Ding et al., 2023)), but there are marginal differences compared with recipes. We use SFT [12] models which trained with different recipes. Specifically, Zephyr-7b-$\beta$'s SFT used the batch size of 512, but 128 was used for the ours SFT model. In addition, regarding the preference dataset, Zephyr-7b-$\beta$ was trained using the original Ultrafeedback (Cui et al., 2023) [13] but we use cleaned version[14]. These changes in training data and the SFT model were aligned with SPA to ensure a fair comparison.

**LLM-as-Judgement.** For LLM-as-judge, we used an SFT model to employ Consitual AI's pairwise comparison prompt for judging preferences (Bai et al., 2022a). Preference is measured by comparing the logprob value of the token output as input to the following prompt (Listing 1). To ensure fair

---

[11] https://huggingface.co/datasets/HuggingFaceH4/ultrachat_200k
[12] https://huggingface.co/alignment-handbook/zephyr-7b-sft-full
[13] https://huggingface.co/datasets/HuggingFaceH4/ultrafeedback_binarized
[14] https://huggingface.co/datasets/argilla/ultrafeedback-binarized-preferences-cleaned

comparison and prevent low judgment performance, evaluation instructions were created using seed preference data which is the same form as Consitual AI's pairwise comparison. (Listing 2) Using these, additional SFT learning is performed to obtain an independent LLM-as-judge model. For this supervised fine-tuning, we set the batch size 32, total epoch is 3, and the learning rate was $2 \times 10^{-5}$. We employed Adam optimizer and a cosine learning rate scheduler with a warm-up phase corresponding to 10% of the total training steps.

**Reward model judgment.** For the reward model baseline, we selected PairRM (Jiang et al., 2023b) due to its high performance on AlpacaEval 2.0 (Snorkel, 2024; Wu et al., 2024). Unlike SPA, which was trained on only 2K gold label data, PairRM was trained on a large-scale dataset. The training data for PairRM includes the following:

- openai/summarize_from_feedback (Stiennon et al., 2020)

- openai/webgpt_comparisons (Nakano et al., 2021)

- Dahoas/synthetic-instruct-gptj-pairwise[15]

- Anthropic/hh-rlhf (Bai et al., 2022a)

- lmsys/chatbot_arena_conversations (Zheng et al., 2023)

- openbmb/UltraFeedback (Cui et al., 2023)

The total number of pairwise samples in this training data is approximately 500K, compared to 2K for SPA. Specifically, the summarize_from_feedback dataset contributes 179K samples, and the hh-rlhf dataset contributes 161K samples, making up a significant portion of the total.

## B.3 Adjusted experimental setups for different LLMs

In Table 5, we conduct the experiments with different LLMs. As they exhibit different characteristics from the difference in backbone and sizes, we slightly adjusted the experimental setups while keeping most identical to the setups in Section 5.1.

**Phi-2.** We slightly adjust the learning rate to accommodate the different characteristics of the Phi-2 ($5 \times 10^{-6}$). In addition, due to the smaller size of the Phi-2, we observe that performance improvements were not evident beyond iteration 2. Therefore, we present the results of iteration 1.

**LLaMA-3 and Phi-3.** We slightly adjust the learning rate to accommodate the different characteristics of models ($1 \times 10^{-5}$). We conduct 1 epoch for the initial DPO training and maintain $\beta = 0.01$ throughout the entire training process. Since performance improvement has been only observed up to iteration 2 in Section 5.2. we conduct the experiments up to iteration 2 for these models.

## B.4 Implementation details

**Resources and computation cost.** For all experiments, we utilized 4 A6000 GPUs. Under this computational resource, generating responses for 10K prompts takes approximately 1 to 2 hour, and preference judging for generated responses also takes about 1 to 2 hour. For training of model with Eq. 10, it takes about 1 to 2 hours per epoch. Therefore, the total time required to complete response generation, preference judgment, and one epoch of training was between 5 to 6 hours per 10K prompt.

**Response generation.** To mitigate the length bias from training with Direct Policy Optimization (DPO), we restricted the maximum token length for self-generated responses to 300 tokens. This limit corresponds to approximately 900 characters.

**Refinement.** Using the simple moving average (SMA) method, we measured the confidence over the most recent 50 samples. Based on this, we calculated the boundary corresponding to the lower K% of the confidence distribution (Eq. 11).

---

[15]https://huggingface.co/datasets/Dahoas/synthetic-instruct-gptj-pairwise

Table 8: **MT-Bench.** Evaluation results on MT-bench with different models. $SPA_{inst}$ and $SPA_{phi}$ denotes the fine-tuned Mistral-7B-Instruct-v0.1 and Phi-2 SFT with SPA, respectively.

| Models | Gold Label (%) | Avg Score |
|---|---|---|
| Mistral-7B-Instruct-v0.1 | - | 6.78 |
| $SPA_{inst}$ (Ours) | - | **7.12** |
| Phi-2 SFT | - | 5.35 |
| Phi-2 DPO | 3.3 | 6.16 |
| $SPA_{phi}$ (Ours) | 3.3 | **6.33** |

```
f'''
Consider the following conversation between a human and an assistant:
[prompt]
Please choose the response that is the most helpful, honest, and
↪  harmless.
Options:
(A) [response 1]
(B) [response 2]
The answer is:
'''
```

Listing 1: Prompt used for judging preferences.

```
f'''
Consider the following conversation between a human and an assistant:
[Horses are hybrids of which two animals?]
Please choose the response that is the most helpful, honest, and
↪  harmless.
Options:
(A) [Horses, scientifically known as Equus caballus, are not the result
↪  of hybridization between two other animals. They are considered a
↪  species of their own within the Equidae family, which also includes
↪  other equines such as donkeys, mules, and zebras. Horses are
↪  believed to have evolved from a small, multi-toed forest-dwelling
↪  animal known as Eohippus, which lived around 60 million years ago.
↪  Over millions of years, these horses evolved into the large,
↪  single-toed animals we know today.]
(B) [Horses are not hybrids of any other animals. They are a unique
↪  species and are not a product of cross-breeding between different
↪  animals. Horses are domesticated animals and have been bred and
↪  developed over thousands of years by humans for various purposes
↪  such as transportation, riding, and agriculture.]
The answer is: (A)

'''
```

Listing 2: Evaluation instruction.

## C   MORE EVALUATION RESULTS IN MT-BENCH

In this section, we further provide the additional evaluation results on MT-Bench (Zheng et al., 2023), which couldn't be presented in the main text, due to the space issue.

We first present (a) task-wise evaluation results and (b) iteration-wise average improvement in Figure 5. As shown in Figure 5a, SPA consistently improves the performance in various tasks. Notably, there is almost no gain in Coding and degradation in Math. We remark that this phenomenon is

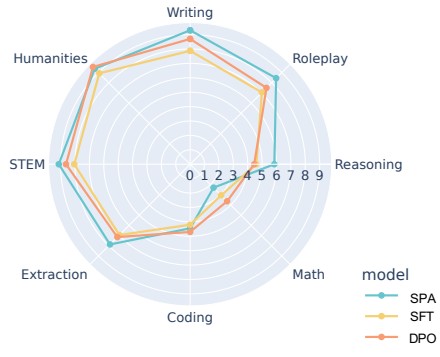

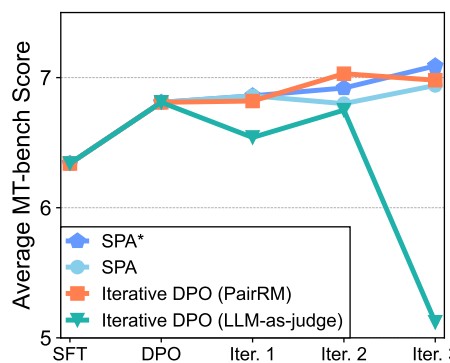

(a) MT-bench task-wise evaluation results

(b) Iteration-wise improvement with MT-bench

Figure 5: **MT-bench Evaluation.** More evaluation results with MT-bench.

Table 9: **Ablation study including MT-Bench.** Evaluation results on AlpacaEval 2.0 and MT-Bench with iteratively trained models (from SFT) under different methodological configurations of SPA. DE, SR, DND are abbreviations of data expansion, self-refinement, and de-coupled noise detection, respectively. The best scores are highlighted with **bold**.

| Methods | DE | SR | DND | AlpacaEval 2.0 | | MT-Bench |
| | | | | Len-control. Win Rate (%) | Win Rate vs. GPT-4 (%) | Avg. Score (0-10) |
|---|---|---|---|---|---|---|
| SFT | - | - | - | 7.58 | 4.72 | 6.34 |
| DPO | - | - | - | 9.03 | 7.68 | 6.81 |
| SPA (Ours) | ✓ | ✗ | ✗ | 14.41 | 19.91 | 6.86 |
| | ✓ | ✓ | ✗ | 14.7 | 19.94 | **7.09** |
| | ✓ | ✓ | ✓ | **15.39** | **21.13** | 6.94 |

commonly observed in the relevant literature (Lin et al., 2024), which indicates that different training (Wang et al., 2023a) or inference (Wei et al., 2022b) schemes might be necessary to improve the performance in these tasks.

Next, in Figure 5b, one can observe that the average performance on the MT-bench is increased with more iterations. Specifically, while the Iterative DPO using PairRM shows the best performance until iteration 2, SPA* (without DND) outperforms it in iteration 3. It demonstrates the effectiveness of our framework for iteratively improving the alignment of LLM.

In addition, we measure the performances of Phi-2 variants and Mistral-7B-Instruct-v0.1 variants on MT-Bench in Table 8; these models are presented in Table 5 and Figure 4, respectively. As one can see, SPA consistently yields the improvement across different backbones of Mistral-7B-Instruct-v0.1 and Phi-2. Lastly, we present the full results of the ablation study (presented in Table 6) that includes the evaluation results on MT-Bench, in Table 9.

# D   MORE QUANTITATIVE RESULTS

In this section, we present more quantitative results to demonstrate the effectiveness of SPA.

**Mitigating length bias with SPA.** Here, we provide a discussion of the relevant experimental results about the length bias present in SPA. During the experiments, we observe that LLMs trained with SPA tend to generate longer responses (see 10), which could be dispreferred depending on the user. Regarding this, we first emphasize that the improvement with SPA is not merely due to longer outputs, as shown by the significant gains in the length-controlled win rate in all experiments in Section 5.

Nevertheless, to further address the concerns regarding this issue, we further investigate whether previously researched length control techniques can be easily integrated into SPA. Specifically, we

Table 10: **SPA with length regularization.** Evaluation results on AlpacaEval2.0 with different variants of Mistral-7B-v0.1 from SPA and the additional length regularization term.

| Models | Gold Label (%) | Len-control. Win Rate | Win Rate vs. GPT-4 | Avg. len (# chars) |
|---|---|---|---|---|
| Mistral-7B-v0.1 | - | 0.17 | 0.50 | 5692 |
| SFT | - | 7.58 | 4.72 | 901 |
| DPO | 3.3 | 9.03 | 7.68 | 1802 |
| Zephyr-7b-$\beta$ | 100 | 11.75 | 10.03 | 1552 |
| SPA (Original, Iter. 1) | 3.3 | 11.88 | 12.95 | 2150 |
| SPA (Modified, Iter. 1) | 3.3 | 11.39 | 12.31 | **2013** |
| SPA (Original, Iter. 2) | 3.3 | 16.23 | 19.94 | 2749 |
| SPA (Modified, Iter. 2) | 3.3 | 14.46 | 18.23 | **2448** |

Table 11: **LLM-as-Judgment with model from previous iteration.** Evaluation results on AlpacaEval 2.0 with different variants of Mistral-7B-v0.1.

| Methods | Len-control. Win Rate (%) | Win Rate vs. GPT-4 (%) |
|---|---|---|
| LLM-as-judge (Iter. 1) | 8.88 | 8.01 |
| LLM-as-judge (Iter. 2, orig) | 9.49 | 8.46 |
| LLM-as-judge (Iter. 2, prev. init) | 9.74 | 10.09 |
| SPA (Iter. 2, ours) | **15.46** | **19.91** |

apply the length penalty approach from RLHFlow Dong et al. (2024). This method heuristically reduces the reward model's reward based on the output length (Eq. 13) during preference labeling. We utilze hyperparamter $\alpha$ between 0.001 to 0.01 that minimize the length increase. The results, shown in Table 10, indicate that this modification successfully reduces the average length while largely preserving the performance improvements from SPA. These results demonstrate that SPA can be easily integrated with existing research related to length control.

$$r_{\texttt{penalty}}(x, y) = r(x, y) - \alpha|y| \tag{13}$$

**LLM-as-Judge with the model from previous iteration.** For the LLM-as-Judgement baseline, we used a fixed model fine-tuned specifically for evaluating preferences between responses, as described in Appendix B.2. On the other hand, at each iteration, the proposed SPA utilizes the previous iteration's model, $\pi_{i-1}$, for the direct preference judgment. Therefore, one can be curious how this difference affect to the performance.

To answer this, we conducted new experiments; at the 2nd iteration, the evaluation model was initialized with the resulting model from the 1st iteration and fine-tuned with the constructed preference evaluation dataset using the seed preference data (see details in Appendix B.2). The evaluation results (denoted as *LLM-as-Judge (Iter. 2, prev. init)*) on AlpacaEval 2.0 are presented in Table 11, along with other methods at the 2nd iteration. While this approach yielded improved alignment compared to the fixed model, SPA still significantly outperformed this baseline. This underscores that the superior performance of SPA arises from its novel preference evaluation techniques rather than the specific evaluation model used.

# E  QUALITATIVE EXAMPLES

In this section, we present examples of the responses from the models trained under different methods (SFT, DPO, and SPA), for a qualitative comparison. The results are presented in Figures 6, 7, and 8. As one can see in the examples, the model trained under SPA provides more detailed and informative responses to various questions.

**Prompt:**
Who is Larry Page?

**SFT response:**
Larry Page is the co-founder and CEO of Google, a multinational technology company that provides various services such as search engine, online advertising, and cloud computing. He was born on March 26, 1973, in East Lansing, Michigan, and graduated from the University of Michigan with a degree in computer engineering. Page and Sergey Brin founded Google in 1998, and since then, it has become one of the most successful and influential companies in the world.

**DPO response:**
Larry Page is an American computer scientist and entrepreneur who co-founded Google, one of the world's largest technology companies, with Sergey Brin in 1998. He is the CEO of Alphabet Inc., the parent company of Google, and has been instrumental in the development and growth of the company. Page is known for his innovative ideas and his focus on developing new technologies that can improve people's lives. He has been recognized for his contributions to the field of computer science and has received numerous awards and honors, including the Presidential Medal of Freedom in 2015.

**SPA response:**
Larry Page is an American entrepreneur, computer scientist, and co-founder of Alphabet Inc., the parent company of Google, one of the world's largest technology conglomerates. Born on March 26, 1973, in East Lansing, Michigan, Page grew up in a family of academics and engineers. He earned his Bachelor's degree in Computer Engineering and Master's degree in Computer Science from Stanford University in 1995 and 1998, respectively.

In 1998, Page teamed up with Sergey Brin, his classmate at Stanford, to develop a new search engine called "Backrub," which later evolved into Google. Initially funded by venture capitalists, Google quickly gained popularity for its innovative search algorithm, user-friendly interface, and extensive collection of online content. Page served as the CEO of Google from its inception in 1999 until 2019, overseeing its rapid growth, expansion into new markets, and development of groundbreaking products and services, such as Android, Maps, YouTube, and Cloud Computing.

Under Page's leadership, Google became a dominant player in the tech industry, with a market capitalization of over $1 trillion and a global workforce of over 135,000 employees. In 2019, Page stepped down as CEO of Google and assumed the role of CEO of Alphabet, the holding company that reorganized Google's core businesses and diversified into new areas, including healthcare, artificial intelligence, and sustainability.

Aside from his entrepreneurial achievements, Page is known for his philanthropic efforts, particularly through the Larry Page Foundation, which supports research in renewable energy, education, and medical innovation. He has also been recognized for his contributions to the tech industry, receiving numerous awards, honors, and recognitions, including the Presidential Medal of Freedom, Time Person of the Year, and Forbes' Person of the Year.

Figure 6: **Example of responses in AlpacaEval 2.0.**

---

**Prompt:**
What causes the northern lights

---

**SFT response:**
The northern lights, also known as aurora borealis, are caused by the interaction between charged particles from the sun and the Earth's magnetic field. When the charged particles from the sun enter the Earth's atmosphere, they collide with the gases in the atmosphere, causing them to emit light. The colors of the northern lights are determined by the type of gas that is colliding with the charged particles. Green is the most common color, but other colors such as red, yellow, and blue can also be seen.

---

**DPO response:**
The northern lights, also known as aurora borealis, are caused by the interaction between charged particles from the sun and the Earth's magnetic field and atmosphere.

When the sun emits charged particles, such as electrons and protons, they travel through space towards the Earth. As they approach the Earth, they are deflected by the Earth's magnetic field, which creates a protective bubble around the planet.

When these charged particles enter the Earth's atmosphere, they collide with the gases in the atmosphere, such as oxygen and nitrogen, causing them to become excited and emit light. This light is what we see as the northern lights.

The colors of the northern lights depend on the type of gas that is being excited. For example, oxygen produces green and yellow lights, while nitrogen produces blue and red lights.

The intensity and frequency of the northern lights depend on the activity of the sun, which can vary over time. During periods of high solar activity, the northern lights can be seen more frequently and with greater intensity.

---

**SPA response:**
The northern lights, also known as aurora borealis, are a natural phenomenon caused by the interaction between charged particles from space and the Earth's magnetic field and atmosphere. The primary cause of the northern lights is the solar wind, a stream of charged particles (primarily electrons, protons, and alpha particles) emitted from the Sun's corona and solar atmosphere.

As the solar wind approaches the Earth, it encounters the magnetic field, which deflects and traps some of the charged particles in a region known as the magnetosphere. The magnetosphere extends outward from the Earth's surface to a distance of several hundred thousand kilometers and is shaped like a teardrop, with the magnetic field lines converging towards the polar regions.

When energetic particles from the solar wind enter the magnetosphere, they collide with the atoms and molecules in the Earth's upper atmosphere, primarily in the polar regions, where the magnetic field lines are strongest. These collisions ionize the atmospheric gases, creating a plasma region known as the ionosphere.

The ionization process releases energy in the form of electromagnetic radiation, including ultraviolet, visible, and radio waves. The visible auroral emissions occur when the energetic particles excite the electrons in the ionized gases, causing them to emit photons with specific wavelengths, resulting in the characteristic colors of the northern lights.

The colors of the aurora depend on the type of gas molecule involved in the collisions and the energy level of the excited states. Green and yellow auroras are typically associated with oxygen atoms, while red and blue auroras are caused by nitrogen molecules. Other gases, such as hydrogen, helium, and carbon dioxide, can also contribute to the auroral spectrum, resulting in additional colors like purple, pink, and white.

The intensity and frequency of auroral displays vary depending on solar activity levels, geomagnetic conditions, and seasonal factors. Solar maximum periods, when the Sun's activity is at its highest, tend to produce more frequent and intense auroral displays in both hemispheres. However, even during solar minimum periods, when solar activity is low, occasional auroral events can still occur in the polar regions.

---

Figure 7: **Example of responses in AlpacaEval 2.0.**

**Prompt:**
What type of soil is suitable for cactus?

**SFT response:**
Cactus plants prefer well-draining soil that is slightly alkaline. A mixture of sand, perlite, and peat moss is a good option for cactus soil. The soil should be lightweight and allow excess water to drain quickly. It is also important to avoid using soil that is too rich in nutrients, as cactus plants do not require a lot of fertilizer.

**DPO response:**
Cacti prefer well-draining soil that is slightly alkaline to neutral. The ideal soil for cacti is a mixture of sand, perlite, and peat moss. The sand provides good drainage, while the perlite helps to aerate the soil and improve water retention. The peat moss adds organic matter and helps to retain moisture. The soil should have a pH level between 6.0 and 7.5. Avoid using heavy, clay-based soils, as they can retain too much moisture and lead to root rot. It is also important to ensure that the soil is not too rich in nutrients, as cacti prefer a slightly nutrient-poor soil.

**SPA response:**
Cacti require a well-draining soil mix that is specifically formulated for their unique needs. The ideal soil for cacti is a cactus soil mix, also known as succulent soil or desert soil.

Cactus soil mixes typically contain a blend of coarse-textured, porous materials such as:

1. Coarse sand: This provides excellent drainage and aeration, preventing waterlogging and root rot. Cacti prefer sandy soils with a particle size of 0.5-2.0 mm.

2. Perlite or vermiculite: These lightweight, volcanic minerals improve soil structure, increase water retention capacity, and promote air circulation around the roots. They also help to break up compacted soil and improve drainage.

3. Pumice or expanded clay: Similar to perlite and vermiculite, these materials improve soil porosity and aeration, while also adding nutrients and buffering acidity levels.

4. Peat moss or coconut coir: These organic components provide some moisture retention and nutrient content, but in small quantities (<20%) to avoid excessive water retention and potential fungal growth.

5. Mineral aggregates: Optional additions such as crushed limestone, dolomite, or volcanic ash can further enhance drainage, pH buffering, and nutrient availability for cacti.

The ideal ratio of these components varies depending on the specific species and growing conditions, but a common starting point is a mix of 60-70% sand, 20-30% perlitevermiculitepumice, and 5-10% organic matter. It is recommended to purchase pre-mixed cactus soil blends from reputable nurseries or garden centers, as they often include optimized ratios and additional amendments for optimal cactus growth.

Figure 8: **Example of responses in AlpacaEval 2.0.**

