# OpenReview forum: "Spread Preference Annotation: Direct Preference Judgment for Efficient LLM Alignment"
_ICLR.cc/2025/Conference — ICLR 2025 Oral_

### Official Review · Reviewer_dZTX · 2024-10-29

**Soundness:** 4
**Presentation:** 4
**Contribution:** 3
**Rating:** 8
**Confidence:** 3

**Summary:**

This paper proposes SPA, an approach to enhance the alignment performance of large language models by using minimal human-annotated preference data.
Introduce a confidence-based refinement of preference labels to reduce the risk of noise in preference learning with generated data.
It is experimentally verified that a tiny percentage of preference data (3.3%) achieves results comparable to or exceeding those obtained using the entire data and the existing optimal baseline method in the AlpacaEval 2.0 evaluation.

**Strengths:**

1. The overall idea is relatively simple, but the method achieves good performance and has great potential with limited human labeling.
2. The paper is well-structured and presents a rigorous methodology, with comprehensive experimental validation that supports the claims made about SPA’s effectiveness.

**Weaknesses:**

This is a very solid piece of work. The proposed method is simple yet effective. I don't have any particular concerns or issues with it.

**Questions:**

1. How do you choose the initial seed data? Are the data randomly chosen as seeds? Are there differences in human preferences across data? How do you handle such differences?
2. In section 4.1, can you elaborate on the sampling process used to generate the responses y1 and y2? In particular, does the model use specific diversity techniques to ensure diversity of reactions, or does it rely purely on randomness in the generation process?
3. How does it ensure the reliability of the expanded data?

---

> ### Author Response · Authors · 2024-11-19
> **Response to Reviewer dZTX**
>
> Dear Reviewer dZTX,
>
> We sincerely appreciate your thoughtful comments. We have carefully considered each of your questions and provide detailed responses below.
>
> ---
>
> **[Q1] How do you choose the initial seed data? Are the data randomly chosen as seeds? Are there differences in human preferences across data? How do you handle such differences?**
>
> The initial seed data is selected through random sampling from the UltraFeedback dataset, which could include diverse human preferences. We do not explicitly address these preference variations, but we observe that SPA remains effective across different random seed selections and seed sizes, as demonstrated in Tables 3 and 4. This robustness stems from SPA’s use of direct preference judgment, which leverages the LLM’s intrinsic knowledge about human preferences derived from the overall seed data, rather than relying on specific data instances like in-context learning approaches.
>
> On the other hand, we’d like to mention that there is a field of research that aims to better capture diverse human preferences in LLM’s responses [1,2]. Since SPA can easily incorporate the modifications in the reward modeling or training objective, we believe that the proposed SPA can be jointly applicable to these existing works.
>
> ---
>
> **[Q2] In section 4.1, can you elaborate on the sampling process used to generate the responses y1 and y2? In particular, does the model use specific diversity techniques to ensure diversity of reactions, or does it rely purely on randomness in the generation process?**
>
> The responses  $y_1$  and $y_2$  are sampled purely based on randomness without applying specific diversity techniques, as described in lines 195-196 of the original draft. However, our framework is flexible and can incorporate various diversity sampling techniques if desired. Given the demonstrated effectiveness of diversity techniques in iterative preference learning (e.g., sampling N>2 responses, then choosing best/worst as the preferred/dispreferred responses [3,4]), we believe that incorporating such methods could further enhance the performance of our framework by enhancing the diversity of the responses.
>
> ---
>
> **[Q3] How does it ensure the reliability of the expanded data?**
>
> First, the proposed direct preference judgment to label the preference between responses (in Section 4.1) yields more reliable data expansion. This is because it leverages intrinsic reward signals directly derived from the target LLM, which are continuously refined through iterative updates. In contrast, prior methods that expand data rely on preference labels generated via implicit prompting or fixed external reward models, limiting their reliability and adaptability.
>
> Additionally, the proposed self-refinement mechanism in Section 4.2 further strengthens the reliability, by addressing labeling noise within the iterative preference learning framework. This self-refinement step uses a logit interpolation to approximate outputs from a more strongly aligned LLM, enabling effective noise detection. By reducing labeling noise in the expanded preference data, this technique enhances the overall reliability and accuracy of the generated data.
>
> We highlight that the enhanced reliability of the expanded data with these components is demonstrated through our experimental results. For example, our SPA approach significantly outperforms prior methods, with improvements such as a 3.52% increase in length-controlled (LC) win rate on AlpacaEval 2.0. Moreover, SPA achieves superior alignment performance using only 3.3% of the seed preference data, compared to DPO method that uses the full 100% seed data.
>
> ---
>
> [1] Zhou et al., Beyond One-preference-for-all: Multi-objective Direct Preference Optimization., arXiv:2310
> [2] Pitis et al., Improving Context-aware Preference Modeling for Language Models., arXiv:2407
> [3] Wu et al., Self-play Preference Optimization for Language Model Alignment., arXiv:2405
> [4] Rosset et al., Direct Nash Optimization: Teaching Language Models to Self-improve with General Preferences., arXiv:2404
>
> ---
>
> If you have any further questions/concerns, please do not hesitate to let us know.
>
> Thank you very much,
> Authors

---

> > ### Comment · Reviewer_dZTX · 2024-11-26
> >
> > Dear authors, thank you for your response! I do not have any additional questions or concerns with this work at this time. Naturally, I will be maintaining my score.

---

> > > ### Author Response · Authors · 2024-11-27
> > >
> > > Dear Reviewer dZTX,
> > >
> > > We are happy to hear that our rebuttal addressed your questions well. Please let us know if you have any further questions.
> > >
> > > Thank you very much.
> > >
> > > Best regards,
> > > Authors

---

> ### Author Response · Authors · 2024-11-25
> **Gentle reminder: The interactive discussion period will end in less than two days**
>
> Dear Reviewer dZTX,
>
> Thank you again for your time and efforts in reviewing our paper.
>
> As the discussion period draws close, we kindly remind you that two days remain for further comments or questions. We would appreciate the opportunity to address any additional concerns you may have before the discussion phase ends.
>
> Thank you very much.
>
> Best regards,
> Authors

---

### Official Review · Reviewer_NqES · 2024-10-30

**Soundness:** 3
**Presentation:** 3
**Contribution:** 3
**Rating:** 8
**Confidence:** 4

**Summary:**

This paper introduces an efficient LLM alignment method, namely Spread Preference Annotation, aiming at reducing the demand for human-labeled preference data. By leveraging the inherent preference of current aligned model, this work generates new preference data points and conducts alignment iteratively. To reduce the potential noise caused by distribution shift, this work incorporates a self-refinement mechanism on preference labels, where this approximate a more strongly aligned model to better identify noise through a linearly extrapolated prediction method. Through experiments this paper proves that SPA can achieve better alignment performance with much less data.

**Strengths:**

1. This work inherits the idea of self-rewarding, but leverages the inherent preference of current aligned model in an intuitive way, which sounds novel to me.
2. The reduction on data usage seems promising, and the performance is robust.

**Weaknesses:**

1. 'De-coupled noise preference detection' is not stated clearly enough in section 4.2. Based on my understanding, $z_{\tilde{\theta}}$ is used to substitute for $z_{\theta}$ in the 'Self-refinement' part, which is also supported in Algorithm 1. If I am correct, I think it would be easier for readers to understand if the final usage of the approximated logits and labels are stated in the main text.
2. Lacks some explanatory discussion on why this method can work on such a small subsets and even perform better than DPO with the full dataset (Details in questions part).

If the author can address my concern in weakness/questions and provide some insightful discussion, I am willing to raise my score.

**Questions:**

1. What is the model used for the 'LLM-as-judge' method in Table 2 ? Have you tried using ${\pi}_{i-1}$ in this baseline?
2. There are lines out of the border in references and appendix.
3. (corresponding to weakness 2) Why this method can work with only 3.3% of total data? Does this method elicit the latent human preference knowledge from pre-trained model, or the 3.3% subset is already enough to define the human preference in UltraFeedback dataset? (Note that this is not a fatal question, feel free to provide any discussion or hypothesis).

---

> ### Author Response · Authors · 2024-11-19
> **Response to Reviewer NqES (1/2)**
>
> Dear Reviewer NqES,
>
> We sincerely appreciate your thoughtful comments. We have carefully considered each of your questions and provide detailed responses below.
>
> ---
>
> **[W1] 'De-coupled noise preference detection' is not stated clearly enough in section 4.2. Based on my understanding, zθ~ is used to substitute for  zθ in the 'Self-refinement' part, which is also supported in Algorithm 1. If I am correct, it would be easier for readers to understand if the final usage of the approximated logits and labels are stated in the main text.**
>
> You’re correct; $z_{\widetilde{\theta}}$ (Eq. 11) is used to substitute for $z_{\theta}$ in the self-refinement step (Eq. 10). Following your suggestion, we have explicitly stated this in Section 4.2 (lines 256-257 in the revised draft) for clarity.
>
> ---
>
> **[W2, Q3] Lacks some explanatory discussion on why this method can work on such a small subsets and even perform better than DPO with the full dataset. Does this method elicit the latent human preference knowledge from pre-trained model, or the 3.3% subset is already enough to define the human preference in UltraFeedback dataset?**
>
> Thank you for the insightful question. As noted in your first conjecture, SPA is indeed able to elicit latent preference knowledge from the pre-trained LLM, allowing it to perform effectively even with a limited number (e.g., 3.3%) of the labeled preference data. Specifically, this effectiveness is achieved through two main techniques in SPA:
> - Direct Preference Judgment (Section 4.1) enables efficient and reliable data expansion, by leveraging intrinsic reward signals directly derived from the target LLM, which are continuously refined through iterative updates.
> - Self-Refinement Mechanism (Section 4.2) further enhances the reliability of expanded data, by addressing labeling noise through the iterative preference learning. By reducing labeling noise in the expanded preference data, it enhances the overall reliability and accuracy of the generated data.
>
> We also highlight that our experiments support this claim well:
> - SPA without seed data (Figure 4 and Table 8): it is assumed that there is no seed preference data (i.e., 0%), and the instruction-tuned LLM is directly used to generate preference data without initial DPO step (1st line in Algorithm 1). Here,  the proposed SPA continuously improves the alignment performance, which supports the effectiveness from the elicitation
> - SPA with varying number of seed data (Table 3): SPA is applied with varying number of seed preference data (0.8% to 10%), and SPA is consistently effective and the improvement grows with increased seed data. It opposes the second part of the conjecture as it implies that the 3.3% subset does not fully capture the UltraFeedback dataset’s human preference knowledge. Remarkably, SPA yields better alignment performance than DPO with full data, if the given seed data is sufficient (e.g., >= 1.7%) to provide the effective guidance to elicit LLM’s intrinsic knowledge about human preference.
>
> Overall, these results indicate that the effectiveness of SPA is from eliciting LLM’s intrinsic knowledge about human preference rather than learning the given seed preference data well.

---

> ### Author Response · Authors · 2024-11-19
> **Response to Reviewer NqES (2/2)**
>
> **[Q1] What is the model used for the 'LLM-as-judge' method in Table 2 ? Have you tried using πi−1 in this baseline?**
>
> For the ‘LLM-as-Judge’ method in Table 2, we used a fixed model fine-tuned specifically for evaluating preferences between responses. Specifically, the training dataset for this model was constructed by converting seed preference data into pairwise comparison prompts, following approaches similar to [1,2]. Then, the common SFT model (used for initialization in other baselines like DPO and SPA) was further fine-tuned on this dataset via supervised learning. More details, such as the training process and the pairwise comparison prompts, are included in Appendix B.2 of the original draft.
>
> Regarding the use of  $\pi_{i-1}$ in this baseline, we conducted new experiments; at the 2nd iteration, the evaluation model was initialized with the resulting model from the 1st iteration and fine-tuned with the same preference evaluation dataset. The evaluation results, as denoted by LLM-as-Judge (Iter. 2, prev. init), on AlpacaEval 2.0 are presented below along with other methods at the 2nd iteration. While this approach yielded somewhat improved alignment compared to the fixed model, SPA still significantly outperformed this baseline. This underscores that SPA’s superior performance arises from its novel preference evaluation techniques rather than the specific evaluation model used. We have added these results and the corresponding discussion to Appendix D of the revised draft.
>
> \begin{array}{c|ccc}
> \hline
> \text{AlpacaEval 2.0} & \text{LC Win Rate (\\%)} &\text{Win Rate (\\%)}
> \newline \hline
> \text{LLM-as-judge (Iter. 1)} & 8.88 & 8.01 \newline  \hline
> \text{LLM-as-judge (Iter. 2, orig)} & 9.49 & 8.46 \newline
> \text{LLM-as-judge (Iter. 2, prev. init)} & 9.74 & 10.09 \newline  \hline
> \text{SPA (Iter. 2, ours)} & \textbf{15.46} & \textbf{19.91} \newline \hline
> \end{array}
>
> ---
>
> **[Q2] There are lines out of the border in references and appendix.**
>
> Thank you for the careful reading and pointing out the editorial problems. We have corrected these in the revised draft (p.13 and p.15).
>
> ---
>
> [1] Bai et al., Constitutional AI: Harmlessness from AI Feedback., Anthropic 2022
> [2] Lee et al., Aligning Large Language Models by On-Policy Self-Judgment., ACL 2024
>
> ---
>
> If you have any further questions/concerns, please do not hesitate to let us know.
>
> Thank you very much,
> Authors

---

> ### Author Response · Authors · 2024-11-25
> **Gentle reminder: The interactive discussion period will end in less than two days**
>
> Dear Reviewer NqES,
>
> Thank you again for your time and efforts in reviewing our paper.
>
> As the discussion period draws close, we kindly remind you that two days remain for further comments or questions. We would appreciate the opportunity to address any additional concerns you may have before the discussion phase ends.
>
> Thank you very much.
>
> Best regards,
> Authors

---

> > ### Comment · Reviewer_NqES · 2024-11-25
> > **Response to author replies.**
> >
> > Thanks for your replies which addresses most of my concerns. As a result, I raise my score to 8 as stated in my review.

---

> ### Author Response · Authors · 2024-11-25
>
> Dear Reviewer NqES,
>
> We are glad to hear that we have addressed most of your concerns. Also, thank you for raising the score!
>
> Please don't hesitate if you have any further questions.
>
> Best regards,
> Authors

---

### Official Review · Reviewer_BPdQ · 2024-11-04

**Soundness:** 4
**Presentation:** 4
**Contribution:** 3
**Rating:** 10
**Confidence:** 4

**Summary:**

This work proposes SPA, a framework to lower the high costs of collecting large preference datasets for alignment. SPA uses an LLM's logits to generate pairwise preference data, without reward model learning or in-context learning, which is then used for preference learning for LLMs. The authors show the practical usefulness of SPA by aligning mistral 7B on 3.3% of Ultrafeedback preference data to achieve strong performance compared to state-of-the-art methods on AlpacaEval 2.0 and MT-Bench

**Strengths:**

1. The paper is well-written and very easy to follow throughout. The paper contextualizes itself within the alignment literature well, covering the fundamentals of pairwise preference learning (Bradley-Terry modeling) to direct alignment algorithms like DPO (Section 3)
2. SPA is able to use only a small seed preference dataset to then directly score preference labels using the implicit reward model learned by DPO (Section 4.1). Since these predictions can be noisy, the authors introduce a novel self-refinement denoising technique using a confidence prediction (eq 9) to smooth the preference label (eq 10)
3. Reproducibility: the authors provide implementation details and hyperparameters in Section 5 (*L311-321*). The authors will open-source the code and models after acceptance, which is appreciated. Lastly, because the modification to the DPO objective is minimal, the authors mention only a few lines of change to the DPO codebase, which is another advantage for practical utility of SPA
4. The authors compare to popular categories of baselines: iterative DPO methods, LLM-as-judge methods, and explicit reward-modeling + RLHF methods (*L288 - L291*) and achieve strong results on AlpacaEval 2.0 and MT-Bench
5. The authors show SPA extends beyond Mistral to other popular LLMs like Phi and Llama (Table 5) and is robust, in the win rate variance sense, to the seed of the initial preference data (Table 4)

**Weaknesses:**

No major weaknesses, mainly minor clarifications:
1. Can the authors provide a little more description about the "length control" aspect of AlpacaEval 2.0 in the main paper? This setting is used in nearly all results, but is not explained clearly where first introduced (Section 5.2)
2. What is "gold label" (Table 1, 5)? Is this the Ultrafeedback preference data? Please make this explicit in the writeup

**Questions:**

Minor Typos
1. "Lengh-control" in Figure 2
2. "additional codes" , L267

---

> ### Author Response · Authors · 2024-11-19
> **Response to Reviewer BPdQ**
>
> Dear Reviewer BPdQ,
>
> We sincerely appreciate your thoughtful comments. We have carefully considered each of your questions and provide detailed responses below.
>
> ---
>
>  **[W1] Can the authors provide a little more description about the "length control" aspect of AlpacaEval 2.0 in the main paper? This setting is used in nearly all results, but is not explained clearly where first introduced (Section 5.2)**
>
> Thank you for the constructive feedback. The length-controlled (LC) win rate in AlpacaEval 2.0 [1] is a newly introduced evaluation metric designed to reduce bias toward longer responses when using LLMs as judges [2,3]. To achieve this, a regression model is trained to separate the contributions of response length and quality, based on data from leaderboard submissions. The LC win rate then estimates win rates by neutralizing the effect of response length, focusing purely on quality. As demonstrated in [1], this metric correlates more closely with human evaluation [4] than the standard win rate. We have added these details about this “length control” aspect of AlpacaEval 2.0 in the revised draft (lines 304-305).
>
> ---
>
> **[W2] What is "gold label" (Table 1, 5)? Is this the Ultrafeedback preference data? Please make this explicit in the writeup**
>
> The term “gold label” refers to the “ground-truth preference label” provided by the UltraFeedback dataset. In our experiments, we use the UltraFeedback dataset in two distinct ways. First, a portion serves as the seed preference data, $\mathcal{D} = \{(x, y_l, y_w)\}$, directly using its ground-truth labels. Second, another portion of the dataset is used solely as a prompt set, $X_i = \{x\}$, by discarding the original labels $y_l$ and $y_w$ to allow to use it for new data generation at each iteration; thus, this portion does not utilize “gold labels.” Therefore, in Tables 1 and 5, we denote “gold label” to clarify the amount of ground-truth preference data utilized, ensuring a fair comparison between methods that rely on different quantities of labeled data. We explicitly mention this in the revised draft (lines 294-295).
>
> ---
>
> **[Q1] Typos.**
>
> Thank you for the careful reading and pointing out the typo! We have corrected this in the revised draft.
>
> ---
>
> [1] Dubois et al., Length-Controlled AlpacaEval: A Simple Way to Debias Automatic Evaluators., arXiv:24.04
> [2] Wang et al., How Far Can Camels Go? Exploring the State of Instruction Tuning on Open Resources., NeurIPS 2023 Datasets and Benchmarks Track
> [3] Zheng et al., Judging LLM-as-a-Judge with MT-Bench and Chatbot Arena., NeurIPS 2023 Datasets and Benchmarks Track
> [4] Chiang et al., Chatbot Arena: An Open Platform for Evaluating LLMs by Human Preference., arXiv:24.03
>
> ---
>
> If you have any further questions/concerns, please do not hesitate to let us know.
>
> Thank you very much,
> Authors

---

> > ### Comment · Reviewer_BPdQ · 2024-11-19
> >
> > Dear authors, thank you for your response! The added information about length control and gold labels addresses all of my (minor) readability related concerns with the manuscript; I do not have any additional questions or concerns with this work at this time. Naturally, I will be maintaining my score.

---

> > > ### Author Response · Authors · 2024-11-21
> > >
> > > Dear Reviewer BPdQ,
> > >
> > > We are happy to hear that our rebuttal addressed your concerns well. Also, we appreciate your support for our work. If you have any further questions or suggestions, please do not hesitate to let us know.
> > >
> > > Best regards,
> > > Authors

---

### Author Response · Authors · 2024-11-19
**General Response**

Dear reviewers and AC,

We sincerely appreciate your valuable time and effort spent reviewing our manuscript.

As reviewers highlighted, we propose a simple (dZTX), yet novel (NqES, BPdQ) method for LLM alignment, that shows strong empirical results (all reviewers) on the comprehensive experiments (dZTX) with clear writing (dZTX, BPdQ).

We appreciate your constructive feedback on our manuscript. In response to the comments, we have carefully revised and enhanced the manuscript as follows:

- Detailed explanation for easier understanding of refinement (Section 4.2)
- Definition of gold label in Tables 1 and 5, and details about length-controlled win rate (Section 5.1)
- Removing lines out of the border (References and Appendix)
- New experiments and corresponding discussions regarding LLM-as-judge using previous iteration’s model (Appendix D and Tabel 11)
- Resolving typos (Figure 2 and line 267)

In the revised manuscript, these updates are temporarily highlighted in $\text{\color{blue}blue}$ for your convenience to check.

We sincerely believe that these updates may help us better deliver the benefits of the proposed SPA to the ICLR community.

Thank you very much,
Authors.

---

### Meta-Review · Area_Chair_thUJ · 2024-12-19

**Metareview:**

This work presents a novel framework, called Spread Preference Annotation with direct preference judgment (SPA), aimed at reducing the high costs associated with collecting large preference datasets for alignment. Overall, all reviewers agreed that this work is novel and important, and they gave very high scores to it. I believe this approach is both simple and effective and makes clearly contribution to LLM Alignment. The achieved results are promising, for example, with only 3.3% of the ground-truth labels this method still achieved superior alignment performance. Based on these, I would like to suggest accept this work.

**Additional Comments On Reviewer Discussion:**

The reviewers pointed out some minor issues, such as data and formatting. In the rebuttal, the authors have addressed the concerns of the reviewers, and all reviewers are satisfied with it. Reviewer BPdQ even raised his/her score to 10 to appreciate the novelty of this work. The reviewers are all positive towards this work.

---

### Decision · Program_Chairs · 2025-01-22

Accept (Oral)